# Investigating the Influence of Feature Sources for Malicious Website Detection

**Ahmad Chaiban [1],\*, Dušan Sovilj [2], Hazem Soliman [2], Geoff Salmon [2] and Xiaodong Lin [1]**

[1] School of Computer Science, University of Guelph, 50 Stone Rd E, Guelph, ON N1G 2W1, Canada; xlin08@uoguelph.ca

[2] Arctic Wolf Networks, 455 Phillip St #100, Waterloo, ON N2L 3X2, Canada; dusan.sovilj@arcticwolf.com (D.S.); hazem.soliman@arcticwolf.com (H.S.); geoff.salmon@arcticwolf.com (G.S.)

\* Correspondence: achaiban@uoguelph.ca

**Abstract:** Malicious websites in general, and phishing websites in particular, attempt to mimic legitimate websites in order to trick users into trusting them. These websites, often a primary method for credential collection, pose a severe threat to large enterprises. Credential collection enables malicious actors to infiltrate enterprise systems without triggering the usual alarms. Therefore, there is a vital need to gain deep insights into the statistical features of these websites that enable Machine Learning (ML) models to classify them from their benign counterparts. Our objective in this paper is to provide this necessary investigation, more specifically, our contribution is to observe and evaluate combinations of feature sources that have not been studied in the existing literature— primarily involving embeddings extracted with Transformer-type neural networks. The second contribution is a new dataset for this problem, GAWAIN, constructed in a way that offers other researchers not only access to data, but our whole data acquisition and processing pipeline. The experiments on our new GAWAIN dataset show that the classification problem is much harder than reported in other studies—we are able to obtain around 84% in terms of test accuracy. For individual feature contributions, the most relevant ones are coming from URL embeddings, indicating that this additional step in the processing pipeline is needed in order to improve predictions. A surprising outcome of the investigation is lack of content-related features (HTML, JavaScript) from the top-10 list. When comparing the prediction outcomes between models trained on commonly used features in the literature versus embedding-related features, the gain with embeddings is slightly above 1% in terms of test accuracy. However, we argue that even this somewhat small increase can play a significant role in detecting malicious websites, and thus these types of feature categories are worth investigating further.

**Keywords:** malicious website detection; phishing; cybersecurity; machine learning; feature extraction

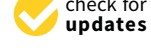



## 1. Introduction

Malicious websites are often designed to host unsolicited content, such as adware, back doors, exploits, and phishing in order to deceive users on multiple levels, and cause the loss of billions yearly according to the IC3 2020 report [1]. It therefore becomes essential to develop robust techniques in order to track and detect malicious websites. The most common form of detection in the field is through configuring Deny/Allow lists, which fails to adapt to new URLs and requires excessive manual attendance. Another method involves utilizing an Intrusion Detection/Prevention System (IDS/IPS), which also falls short on several fronts [2]. In order to combat those drawbacks, more robust methods that employ Machine Learning (ML) techniques have been studied. These include training a multitude of ML classifiers using a standard feature-based approach [3–9], the usage of transformers and CNNs on raw URLs [10], the anti-phishing system that employs NLP-based features [11], and ML-based solutions against typo-squatting attacks [12].

The information necessary for these processes is generally extracted in the form of three feature types/categories: lexical, host-based, and content-based. Lexical features tackle the natural language processing (NLP) aspects of the URL and quantify them as statistical properties. Host-based features tackle properties such as the IP address, domain name, WHOIS information, connection speeds, and location. Content-based features include JavaScript, HTML, and images derived from or used to build websites. In terms of application, several studies have utilized these types of features, either focusing on a single type [11,13], or utilizing a mixture of the different feature categories [14].

A survey paper [2] discusses various ML techniques that have been attempted for the malicious URL detection and what type of data has been used in this domain. All the studies considered have been solely focused on a single aspect of the malicious website—whether it be URL as raw text, host- and content-based features extracted from various sources, or images—multiple images within the website or a screen-capture of the website's front page. Another study in [10] mainly focuses on using transformers on raw URLs, with comparisons made to the traditional feature-based approach for website classification. Even though a wide range of models is covered, the study does not touch upon how each of those feature categories influence the outcome of the trained models. Our aim in this study is to investigate how these different feature sources interact in unity and what are the most relevant contributions from each feature category.

In order to facilitate the main objective of this research study, we required a dataset that is representative of population-level data compiled from distinct feature sources. Due to the nature of security data, certain studies prefer not to disclose their datasets due to privacy concerns, which therefore limited the first phase of this study to open-source datasets. Moreover, many of the available datasets did not provide a broader collection of feature types. We were eventually able to perform a broader analysis of over 25 datasets, and only one of them satisfied our initial feature category constraints. That one dataset was analyzed with more depth and used for ML training. However, it had its shortcomings when dealing with certain feature types, which required proper analysis and a reduction of bias in data that is to be used.

To appropriately evaluate the influence of these feature types found in the literature, we constructed a new dataset—named GAWAIN—which includes both the features themselves and the relevant raw data. The scripts used to construct this data are publicly available [15], allowing researchers to incorporate and extend them into their research. We hope this provides enough incentive for other researchers in the cybersecurity domain to utilize open-source information. Once we built the dataset, a feature type analysis was then performed in order to provide insight into which features contribute the most to this classification problem, and give researchers the best chance for solving it. In particular, the contributions of this research study include the following:

- Conduct an empirical study into the accuracy contributions of the various feature types that can be used for this classification problem. It is worth noting that while other studies have evaluated and combined features extensively, our distinction and contribution stems from the specific sources of the features, which are embeddings from images, URLs, and JavaScript content extracted using the Distilbert, Codebert, and Longformer transformers, and the different PCA and Chi-squared feature selection reductions of those embeddings combined with lexical, host-based, and content-based features typically seen in the literature. This would provide researchers with insight into how each of these features types, and their combinations, can contribute to classifying malicious websites.
- Provide the open-source community with a new dataset GAWAIN in order to add a dataset extracted with the intention to give the community this potentially innovative combination of feature sources to work with in future studies of their own. This dataset was constructed from existing real-world URLs that have been pre-labeled as malicious or benign throughout multiple sources.

It is vital to mention that the dataset would ideally represent population-level data and be free of bias. This is a near impossible scenario to achieve since bias is inevitably inherent in all data. It is worth mentioning however, that our aspiration was to alleviate any apparent biases in the data and repeat the process of extracting the features, which gives us the ability to control the results of our data. Apruzzese et al. discussed this issue at length, performed experiments on datasets of this kind, and attempted to evaluate the efficacy of ML in the Cybersecurity field [16,17]. In their work, they evaluated several datasets and attempted to detect methods of attack that involve large amounts of network traffic, and the results yielded reasonable but not high accuracy. For instance, after deploying their models for intrusion detection, the authors mentioned the need for rigorous dataset creation and training, else the results would otherwise be underwhelming, yielding high false negative rates, and recalls that range between 60% to 70%.

This therefore opened up the opportunity to research into constructing and analyzing datasets that reflect population-level data as much as possible, and assessing ML/feature performance for malicious website detection. This study is motivated by this direction of research, and seeks to make advances in this field through its ML implementation and feature creation on datasets of that nature. It also introduces GAWAIN dataset, which has been added to the open-source community, and may challenge and enable researchers to investigate the malicious website detection problem further.

The paper is organized as follows. Section 2 discusses the literature with the focus on feature types and engineering. Section 3 investigates one particular openly available dataset where we outline several of its shortcomings. In Section 4, we describe a new dataset GAWAIN and all the features. We study the influence of constructed features on the new dataset in Section 5. The concluding remarks are given in Section 6.

## 2. Previous Work

While the literature contains a plethora of ML techniques applied for this task [2,18], we outline the studies that primarily revolve around the design and use of different feature categories which we believe provide a significant enough contribution to our study's design.

Sahingoz et al. built a real-time anti-phishing system that employs seven classification algorithms and NLP-based features [11]. In terms of their data, the authors have constructed a new dataset, as they could not find a suitable open-source dataset, by scraping PhishTank. It contains 73,575 URLs, of which 36,400 are benign URLs and 37,175 are malicious. Their dataset was constructed by an algorithm that extracted lexical-based features from the URLs. The algorithm parsed the URL string for special characters, certain random words and keyword counts to name the few features, and then a "maliciousness" analysis was performed on these word vectors and keywords. Their final results showed the promise of lexical features in the classification problem. Their tests resulted in a Random Forest classifier evaluating their test set at 97.89% accuracy, with other models performing similarly.

Moubayed et al. proposed a solution against typo-squatting attacks [12]. This type of attack refers to the registration of domain names that are similar to existing popular brands, allowing cyber-criminals to redirect users to malicious or suspicious websites. Moubayed et al. constructed an ensemble-based feature selection method in order to construct the final training dataset for their classification problem. This feature selection method performed a correlation-based feature selection, information gain-based feature selection, and a one rule-based feature selection on the data in parallel, which was then followed by a feature reduction technique. A bagging ensemble model was then trained on the dataset. Their experimental results portrayed that the proposed method resulted in a high test accuracy.

Dalgic et al. proposed an image classification approach for malicious website detection [13]. Their idea was to extract several MPEG-7 features and study each one's influence on detection. Only two classifiers were employed in order to assess the five features. The highest F1 score on their testing samples resulted from a Random Forest classifier trained with the Scalable Color Descriptor (SCD) and was 89.5. This study suggested that

there can be potential in using images of web pages, possibly image embeddings or other extracted features.

Shahrivari et al. used a PhishTank dataset, which contained about 11,000 websites divided into malicious and benign [14]. The features of their dataset included content, lexical and host-based features. A correlation matrix was utilized for feature selection, and several machine learning models were trained. Their strongest model tested at 98.26% test accuracy. These results and methodology of grouping features of different types together also seems to suggest the power of combining these features, which further suggested a reason for our detailed study into feature contributions in this classification problem.

Rudd et al. conducted a large study that utilized over 20 million URLs [10]. The study is focused on specific neural network models, transformers and Convolutional Neural Networks (CNN), targeting the raw URL data. With different training regimes and loss functions, they achieved AUC scores of 0.956 and 0.959 for transformer and CNN models, respectively. Due to privacy concerns, the data utilized in their study is not publicly available. However, the results indicate that *embeddings* produced by neural networks from raw URLs perform on par with feature-based methods. This is another avenue we explored in this study—the influence of transformer embeddings alongside other feature categories.

Yi et al. focused on three contributions in their work [19]. They present two feature types for web phishing detection—original features and interaction features. Original features are the direct features of URLs, including special characters and domain age. Interaction features include the interaction between websites, such as the in-degree and out-degree of a URL. They also introduced Deep Belief Networks (DBN) to detect web phishing and discuss the training process of this model with the appropriate parameters. Real IP flow data from ISPs were also utilized to evaluate the effectiveness of the detection model on DBN. The recall method was used for evaluation, yielding a 90% True Positive Rate.

McGahagan et al. conducted a study that performs a similar experiment to our study [20]. They gathered around 46,580 features that were extracted from a response to a web request and used various feature selection techniques to compare the performance of these combinations to those in prior research. They build several unsupervised and supervised ML algorithms for different feature combination/transformation scenarios. Their final results showed that a reduced combination of features was able to achieve more efficient and comparable detection accuracy, while this study is quite similar to ours, we take a combination of features that is more diverse in terms of their sources. We take embeddings from images, URLs and JS code, use various feature selection/reduction techniques to create more combinations from them and combine them with typically used content-based, host-based and lexical-based features.

In summary, these studies provide insight into how certain feature types may perform with different classifiers, and outline the necessity for conducting a study that considers all feature types under one framework. This was made our main objective, that is, to give insight into how different feature types and individual features contribute to the ML-based malicious website detection problem.

## 3. Preliminary Experiments

In this section, we show the detailed investigation performed on one particular dataset after performing a wider search and inspection of over 25 publicly available datasets. The reasoning for this narrow focus are the following check-marks that we set in advance for the study: large enough sample size (preferably more than 100 k), balanced dataset in terms of class labels (if possible), constructed features cover multiple feature categories, and access to raw data. Among the datasets that we were able to find, only the one designed by A.K. Singh [21,22] passed enough check-marks to be considered for the study.

### 3.1. A.K. Singh's Dataset

The selected dataset contains 1.3 million samples and has been updated fairly recently (2019). The main reason we selected this particular dataset is the feature set that covers

lexical, host-based and content-based categories. Moreover, the dataset offers access to raw URLs and the HTML content page of each website in *string* format. A summary of the features offered by this dataset is given in Table 1 (features with * symbol).

**Table 1.** List of features covered in this study. The labels indicate: * part of the A.K. Singh's dataset, † part of the expanded dataset used in Section 3, ⋄ not part of the our final dataset GAWAIN.

| Category | Feature | Description |
|---|---|---|
| Raw | URL of the website | String format of the URL of the website. |
| | HTML page content | The HTML page code of the homepage of the website in string format. |
| | Hostname | The extracted hostname of the website in string format. |
| | Domain name | The extracted domain name of each website in string format. |
| | Label of the website | A binary label that states whether the website is malicious or benign. |
| Lexical | Length of the URL * | The number of characters in the URL. |
| | Number of underscores, semi-colons, subdomains, zeros, spaces, hyphens, @ symbols, queries, ampersands, and equal signs † | The number of each of these characters counted in the URL. Note that all these characters are considered as separate features. |
| | Hostname length † | The number of characters in the hostname extracted from the given website. |
| | Ratio of digits to URL † | The number of digits divided by the length of the URL. |
| | Ratio of digits to hostname † | The number of digits divided by length of hostname. |
| | IP address in URL † | A binary label that states if an IP address exists in the URL. |
| | Existence of @ symbol in URL † | A label that states if the URL contains any @ characters. |
| | Domain length | The number of characters in the extracted domain name. |
| | Unique URL characters, numbers and letters | Taking the total number of unique characters, numbers and letters in a URL. Note that each of these is their own feature. |
| | Ratio of letters to chars | The unique letter count divided by the total unique character count in a URL. |
| | Ratio of numbers to chars | Count of numbers in a URL divided by the total unique character in a URL. |
| | Top-level domain * | The top level domain of a website as a category label. |
| Content | Length of JavaScript code * | Total number of characters of JavaScript code present in HTML content. |
| | Length of deobfuscated JS code *,⋄ | Length of the JavaScript code put through deobfuscation tool. |
| | URL count in content † | The total number of URLs found in the HTML page content [23,24]. |
| | Unique URL count in content † | The unique number of URLs found in the HTML page content [23,24]. |
| | Suspicious JS function count † | The number of suspicious functions found in the JavaScript code [25]. |
| | JavaScript function count † | The total number of JavaScript functions called in the code [23,24]. |
| | Content length | The total number of characters of the HTML content page. |
| | Script tag references | A count of the total number of SCRIPT tags in the HTML page content. |
| | Contains HEX | Binary indicator about webpage's HTML content containing any hexadecimal characters. |
| | HEX length | The number of Hexadecimal characters in the HTML content page. |
| | DCD MPEG-7 † | The 5 dominant colors of an image. They are extracted as 5 different columns. |
| | Average length of JS arrays | The average length of arrays in the JS code present in HTML content. |
| | Maximum length of JS arrays | The maximum length of arrays in the JS code present in HTML content. |
| Host | IP address * | String format of the IP address of the website. |
| | Geographic location | The name of the country the website is hosted in. |
| | WHOIS information* | A binary label that states whether the WHOIS information for the website is complete or incomplete. |
| | HTTPS * | A binary label that states whether the website is secure HTTP protocol or not. |
| | Is in Alexa's top 1 million | A binary label that describes if the given domain name of a URL exists in Alexa's top 1 million domains. |
| Embedding | Image embeddings | The embeddings of the website's image. They are extracted using MobileNetV2 [26]. |
| | Content embeddings | The embeddings of the HTML page content. Extracted using CodeBert [27,28]. |
| | URL embeddings (Distilbert) | URL embeddings extracted using the Distilbert Tranformer [29,30]. |
| | URL embeddings (Longformer) | URL embeddings extracted using the Longformer Tranformer [30,31]. |
| | Mean statistic for embeddings | URL and image embeddings, as well as chi-squared feature selected embedding dimensions, are summarized as a single representative number—the mean of the embedding vector (or parts of the vector). |

### 3.2. Additional Malicious Samples

The original dataset contains over 1.3 million samples, out of which nearly 30 k are malicious cases. To further augment the data, additional malicious websites have been added to the pool for a total of 46,418 samples with malicious label. These additional samples were taken from PhishTank http://phishtank.org/ (accessed on 10 August 2021)

and AA419 http://wiki.aa419.org (accessed on 10 August 2021), which then underwent a feature extraction phase in order to be merged with the original data. A representative batch of benign data equal in size to the malicious cases, 46,418, was selected and brought down the total number of samples in the expanded dataset to 92,836. We have limited the size of benign samples for the purpose of investigation mostly due to computational constraints.

### 3.3. Expanding A.K. Singh's Dataset

Additionally, we included supplemental features found in Table 1 that are found in other studies which can be extracted once we have access to raw data. The aim is to enrich the original dataset and consider as wide range of features as possible. For a description of each feature, refer to Table 1 and † symbol. For each feature or group of features, the source is outlined below.

#### 3.3.1. Lexical Features

Lexical features are based on the raw URL of the website and involve computing statistical properties of the characters plus certain additional variables specific to internet terminology. Examples of these features are: *ratio of digits to letters in URL/hostname*, *URL contains an IP* and *number of subdomains*. This category of features has been used for malicious URL detection [32] and for classifying DNS Type-squatting attacks [12].

#### 3.3.2. Host-Based Features

Host-based features comprise of the characteristics of host-name properties of a raw URL. They typically provide information regarding the host of the webpage. For instance, domain name properties, ports, named servers time to live and connection speed. This category of features has been used for the task of Phishing detection in [14].

#### 3.3.3. Content-Based Features

Any source-code behind the webpages serves as the potential input for content-based features. This can be HTML, JavaScript, CSS, PHP, or any other code. Example features include: *total* and *unique URL count* in the code, *total JavaScript function count*, and *malicious function count*. Studies that have successfully utilized these features tackle detecting malicious web links alongside their attack types [23] and detecting the presence of malicious JavaScript code [24]. The *suspicious JavaScript function count* feature involved counting the number of functions associated with potentially risky JavaScript code [25].

### 3.4. Feature Inspection & Experimental Results

Upon studying the feature contributions in the original and the expanded dataset, we uncovered an anomaly in the JavaScript-based features. It was found that the distribution of values of the data were clearly separated and we concluded that there appears to be bias emanating from these features. Figure 1 outlines this clear malicious-benign separation caused by JavaScript features. It would be quite unrealistic to accept this separation as ground truth, especially for a classification problem of this magnitude and nature. We will be investigating and addressing this issue at a later stage of the study. At this stage, we evaluate several classic ML models on both datasets, with the addition of two more variations which have JavaScript features removed from the initial data. This gives us four dataset variations for investigation purposes:

1. Original data provided in [21];
2. Original data with JavaScript features removed;
3. Expanded data with features and malicious samples described in this section;
4. Expanded data with JavaScript features removed.

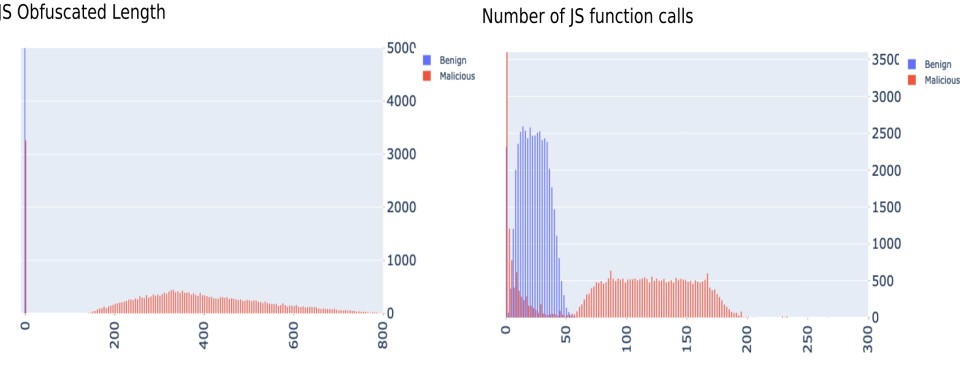

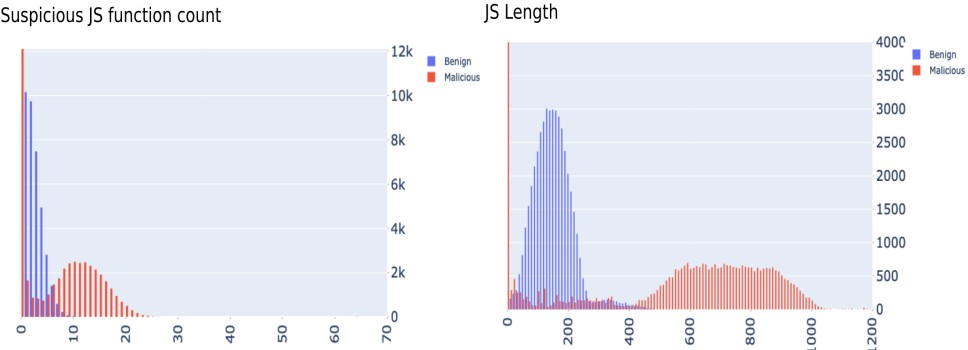

**Figure 1.** JavaScript features distributed by class for A.K. Singh data.

The ML models used for this phase and their results are outlined in Table 2. At first glance, the accuracy scores may seem optimistic. However, after a thorough inspection of the feature space given by A.K. Singh's data, we identified an anomaly in the JavaScript-based features which suggested the existence of bias in the data. In general, data is considered biased if the sampling distributions in the dataset do not reflect population-level distributions, and in this case, it may be considered sample selection bias. This therefore opened up an investigation into JavaScript code in malicious websites, which might further validate these results. The JavaScript code present in the raw information as part of the original data comes in a heavily processed state, compared to what we extracted revisiting the original malicious websites as part of our feature extraction pipeline. This may have skewed the feature extraction phase and elevated the bias to a high degree to render any application of machine learning techniques meaningless.

To further emphasize, there are studies which display more accurate and logical representations of their JavaScript feature sets. Studies [33–36] are of varying times and datasets which plot distributions and t-SNE plots of their JavaScript features, which count deobfuscated code length, number of events and various vector embedding representations. These features are quite similar to the ones considered in this study, which allows us to benchmark the extent of the bias. When this comparison to previous studies is made, we concluded that we would not proceed with the current variations of A.K. Signh's dataset, but move forward with creating our own new dataset based on the same malicious-benign URLs given by A.K. Singh, but extract the features ourselves.

**Table 2.** Classification accuracy (%) with and without JavaScript features for original and expanded A.K. Singh dataset.

| Model | Variation 1 (with JS) | Variation 2 (w/o JS) | Variation 3 (with JS) | Variation 4 (w/o JS) |
|---|---|---|---|---|
| N. Bayes | 93.09 | 83.72 | 85.57 | 55.26 |
| SVM | 96.03 | 85.36 | 98.13 | 86.47 |
| KNN | 94.54 | 82.82 | 96.70 | 81.86 |
| XGBoost | 97.05 | 86.43 | 99.02 | 88.25 |
| AdaBoost | 95.88 | 85.63 | 97.69 | 86.17 |

## 4. Dataset GAWAIN

After detailed investigations into publicly available datasets, and a closer inspection of a dataset with the most promising properties, we found that while there may exist a representative dataset for the problem of detecting malicious websites, we thought it more beneficial to the community, and to our own feature contribution study, to build our own dataset. This allows us to more accurately compare the performance of the feature types with one another. Therefore, we decided to construct a new dataset GAWAIN with as many features utilized across multiple studies, and also take advantage of recent advances in the field. It is important to stress that the features of these websites may change over time, or may not be accessible at all in the long-term, and this is the motivation to allow other researchers access to our data collection and processing pipeline.

In order to build this dataset, a large set of active links was required in order to extract the content, images and other host-based features. Therefore, hundreds of thousands of URLs were taken from the 25 initially collected datasets, as well as the PhishTank http://phishtank.org/ and the AA419.org http://wiki.aa419.org/ databases. The final result was 105,485 samples of data, with 61,080 being benign and 44,405 malicious. The images are snapshots of the website's main page, and are extracted using the Selenium library. Two examples of website snapshots are given in Figure 2.

*Feature List*

The dataset includes all of the features that are part of expanded dataset used in Section 3, as well as several new features as part of host-based and content-based categories. For a full list refer to Table 1. It is also worth mentioning that there was no obfuscated JavaScript (JS) code in the samples. This was verified through the usage of "JS-Auto-DeObfuscator" tool [37] and the methodology from [38,39]. Therefore, this feature was dropped from the dataset.

Here, we highlighted the features that are extracted to supplement features utilized in Section 3:

- JavaScript average and max array length.
- HTML content length.
- Number of JavaScript tag references.
- Number of semicolons, zeros, spaces, hyphens, @ characters, queries, ampersands and equals.
- Domain and domain length.
- Dominant Colors Description (DCD).
- Hex length in content.
- Contains Hex.
- Unique number of characters, numbers, and letters.
- Ratio of letters to characters, numbers to characters.
- Is in Alexa's top 1 million.
- URL, content and image embeddings.

- Mean summary statistic for URL, content and image embeddings.

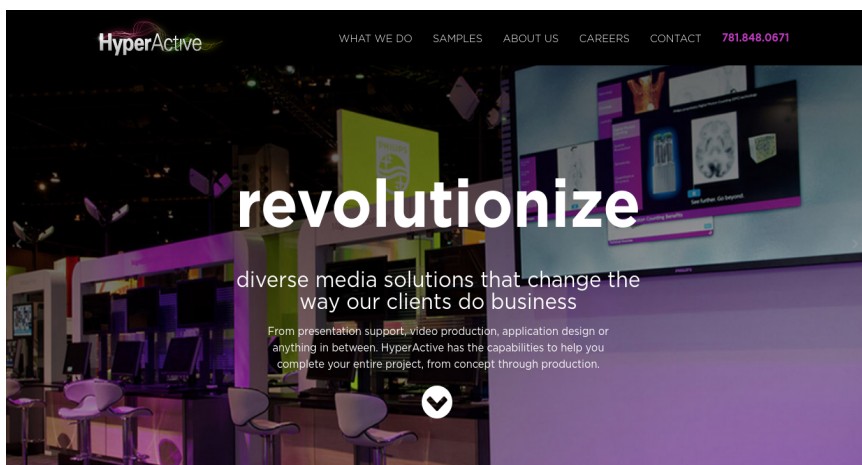

(**a**) Benign labeled image example of a website from Dataset GAWAIN

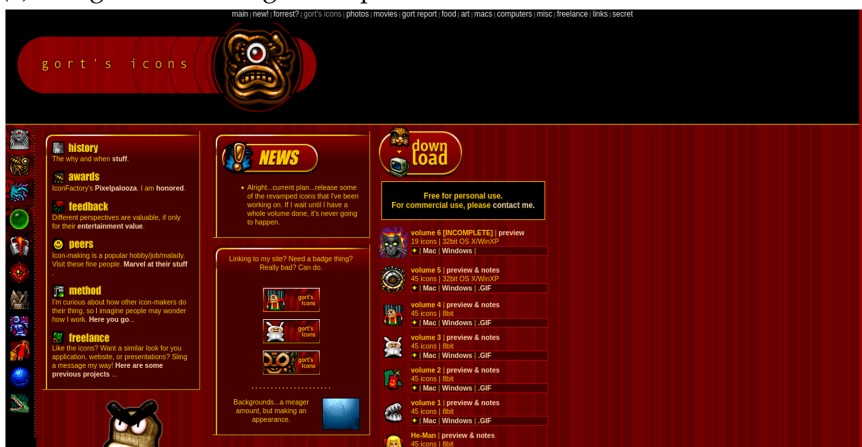

(**b**) Malicious labeled image example of a website from Dataset GAWAIN

**Figure 2.** Examples of two website snapshots.

In terms of the JavaScript bias, after conducting another investigation into the features, and extracting them using scripts from the original paper [21], it was evident that this bias vanished. Figure 3 shows a distribution plot of those JavaScript features by class label for GAWAIN samples. There is almost no separation in this data, which we believe is more realistic scenario to be encountered where attackers try incorporate as many techniques to avoid detection.

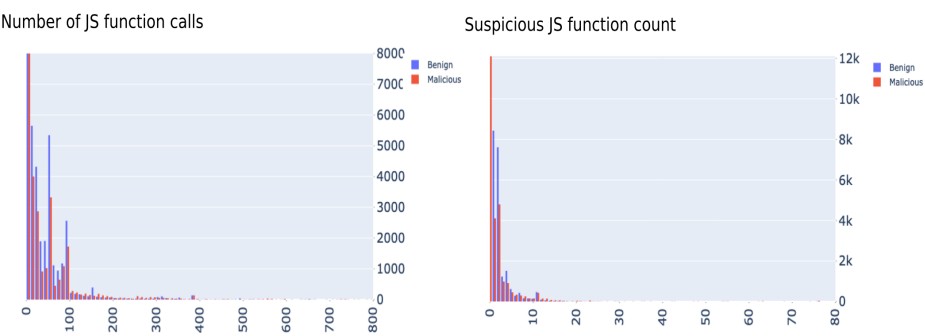

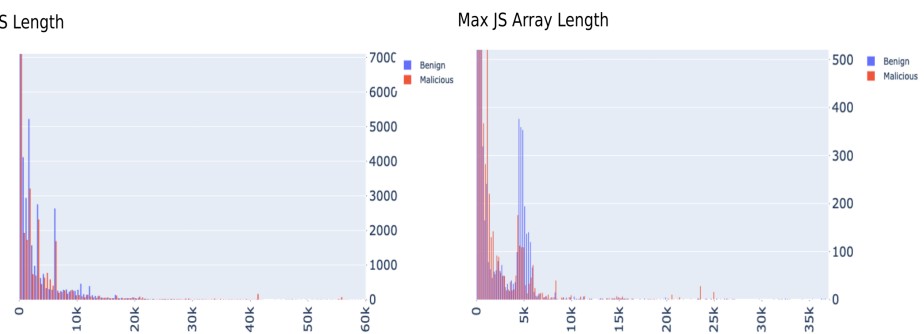

**Figure 3.** GAWAIN's JavaScript distribution graphs.

## 5. Experimental Results—GAWAIN

In this section, we explore the influence of all the constructed features, and their interplay in determining website maliciousness. Our main interest point is how features from different categories affect the outcome of classification models and which are the most contributing factors.

### 5.1. Feature Categorization

Before the ML training phase, the features were divided into several categories in order to measure their contributions. In particular, the features were divided into lexical, host-based, content-based, and all the different variations of the embeddings. In addition, we further expanded the embeddings category with an extra preprocessing step. The original embeddings produced with the neural network models are high-dimensional (with 512 features or more), and we resorted to dimensionality reduction steps to bring that number down. We performed: (1) Principal Component Analysis and reduced the embedding components to 10, 25, and 50 for all the embedding variations, and (2) Chi-squared feature selection targeting the first 20 variables.

### 5.2. Methodology

The feature sets were initially trained and tested using an XGBoost model, which assisted in guiding which features seemed to contribute positively to the test accuracy score. On each category or combination of features, the same train–test split of 90–10% was taken, and passed to the XGBoost classifier. The tuning of the XGBoost model was performed via cross-validation on the training split, with the choice of hyperparameter values given in Table 3 where the final selection of values is highlighted.

After training the model on each feature category and combinations of feature categories, and finding the final best two results, the rest of the models, an SVM, Naive Bayes, Adaboost, and a KNN model, were trained on those final results with the same 90–10% train–test split. The results of these classifiers seemed to under-perform, with accuracy

scores ranging between 60% to 70%, with the exception of a Random Forest Classifier which performed in the low 80% range, almost matching XGBoost.

**Table 3.** XGBoost hyperparameters and tested values.

| Parameter Name | Assigned Value |
| --- | --- |
| Max Depth | {3, 4, 5, 6, **7**, 8, 9, 10} |
| Minimum child weight | {**1**, 5, 10} |
| Number of estimators | {100, 120, 150, **165**, 180, 200, 300, 1000} |
| Colsample by tree | {0.6, 0.8, **1.0**} |
| Learning rate | {0.1, 0.2, **0.3**, 0.4, 0.5} |
| Tree method | Auto, **Exact**, Approx, Hist, GPU-hist |
| Booster | Gbtree, Gblinear, **Dart** |
| Gamma | {1, $10^{-1}$, $10^{-2}$, $10^{-4}$, $10^{-6}$, $10^{-8}$, $\mathbf{10^{-10}}$} |

### 5.3. Model Performance

The final results of training on each feature category with XGBoost and the rest of the models can be found in Table 4. Note that the feature categories are described in Table 1. We also designated a combination of lexical, host-based and content-based features as *Baseline* category. Most of these features are part of other studies, and therefore their unity should carry the most relevant information for the problem. The accuracy of these feature sets was evaluated in two ways. The first is Individual accuracy, which is the test accuracy after training XGBoost on each feature category separately. The second is combined accuracy, which is the test accuracy after training on each feature category combined with the preceding feature category in Table 4. For example, under the content embeddings—PCA 10, that feature is studied in combination with Image embeddings—PCA (the best performing highlighted feature), URL embeddings—PCA 10 (for the same reason mentioned earlier), and the baseline.

The combination of features follows a cascade style additions, that is, once we have the established the best performance for feature categories $f_1, \ldots, f_k$ together, we proceed to add the next category $f_{k+1}$. For example, in Table 4, the *Host-based* combined accuracy refers to combination of *Lexical + Host-based* features, then *Content-based* features are added on top of *Lexical + Host-based* combination. The Longformer results are a batch of experiments where each one is added separately on top of *Baseline* combination, with the highlighted score selected as the representative score for the batch, and in this case it is *PCA 10* variation. This score *URL embeddings [Longformer] PCA 10* is added to the *Baseline* before moving to the next batch of scores involving image embeddings. The process is repeated until we have no feature categories left, with the random order of embeddings to add.

Inspecting the Individual accuracy result in Table 4, the top feature categories are the *URL embeddings* (both Longformer and Distilbert) and *Content-based* with test accuracy of more than 76%. We see that from two distinct sources we are able to achieve the similar score, where *Content-based* features should contain the necessary information for more precise classification, in principle, as opposed to the URL related categories. This does align with certain studies showing the importance of pure URLs as the main information carrier. This is exemplified by the results where the URL embeddings have slightly higher performance compared to the manually engineered *Lexical* features. We also notice that embedding the content itself loses a lot of performance, and that embeddings themselves are not capturing enough context that we are able to extract manually, which is the opposite outcome when comparing against the *Lexical* category.

**Table 4.** Test accuracy for XGBoost across different feature combinations. The individual column shows the accuracy when these features are used for training on their own. The combined column shows accuracy when combined with the preceding feature category in the table. Highlighted scores are the best score for that particular combination of categories.

| Feature Category | Individual | Combined |
|---|---|---|
| Lexical | **75.48** | N/A |
| Host-based | 65.70 | **80.02** |
| Content-based (Baseline) | 76.30 | **83.11** |
| URL embeddings (Longformer) | | |
| all | 76.38 | 83.23 |
| PCA 10 | 75.41 | **83.73** |
| PCA 25 | 75.24 | 82.41 |
| PCA 50 | 75.25 | 82.09 |
| Chi-sq | 76.43 | 83.29 |
| Image embeddings | | |
| all | 60.27 | 82.24 |
| PCA 10 | 62.57 | **83.69** |
| PCA 25 | 61.73 | 83.46 |
| PCA 50 | 61.16 | 82.82 |
| Chi-sq | 61.29 | 83.62 |
| Content embeddings | | |
| all | 67.63 | 81.89 |
| PCA 10 | 67.71 | **83.74** |
| PCA 25 | 68.11 | 83.58 |
| PCA 50 | 67.76 | 83.06 |
| Chi-sq | 66.57 | 83.59 |
| URL embeddings (Distilbert) | | |
| all | 76.62 | 82.47 |
| PCA 10 | 72.97 | **83.79** |
| PCA 25 | 73.24 | 83.57 |
| PCA 50 | 74.08 | 82.99 |
| Chi-sq | 75.12 | 83.64 |
| Best Score (Feat selection) | N/A | **84.27** |

Looking at the combined results, the *Baseline* combination captures most of the information from the features and sets the bar at 83.11%. This number represents the performance of the most commonly used features in other studies combined, and it immediately stands out that the result is quite a margin away from other published results (above 90%). We attribute this to our preprocessing pipeline, where we have showed in earlier sections that we have removed certain type of bias. With newly constructed features, and with one of the most commonly employed models for classification purposes in XGBoost, we were only able to reach 83% indicating that the problem appears to be a lot more difficult.

Considering how we cascaded the feature combinations, the introduction of embeddings on top of the *Baseline* brings little improvements: going from 83.11% to 83.73%, then 83.69%, then 83.69%, and finally to 83.79%. In the end, we have minor improvement that is being added from the embedding components. This suggests that the *Baseline* category already contains enough context that overlaps with what embeddings are extracting from raw data. However, we believe that even this small increment is vital if the end goal is to detect as many malicious websites, potentially preventing future intrusions. We would like to stress again that with this new dataset, the problem has become much harder compared to other studies, and even the minor improvements are worth considering. Finally, we were able to get the best score of 84.27% by manually selecting certain features that are not part of an automated pipeline, but were part of a trial and error approach. In the next part, we point to an interesting result when it comes to individual feature contributions.

*5.4. Feature Contributions*

The insights were at their most valuable when the feature combinations are divided into a combination of lexical, host, and content-based features, with and without including the embeddings. The specific feature importance/contribution for the classification task are outlined in Tables 5–7 for the XGBoost model. We are interested in which features are most important for the *Baseline* category, and how the introduction of different embeddings affects the feature contributions. Since we have two dimensionality reduction methods tested, we present two sets of feature contributions, one for PCA and the other for Chi-square test. The XGBoost model offered insight into which features had been the most contributing factors for segmenting the training data.

Several observations can be made regarding the feature contributions of each of these categories. The simplest observation is that as the embeddings are added to the features, the contribution to the classification is split more equally. We observe that URL embeddings play a major role and replace certain Lexical features altogether, and this replacement contributes slightly to the overall test accuracy shown in Table 4. There are many possible reasons for this change. One could be that the embeddings are compensating for other characteristics of the URLs and content that the other features may not be highlighting. These feature contributions may be a better justification than test set accuracy for the usage of the embeddings alongside the extracted features, since they provide a deeper explanation for the influence of these embeddings on the ML. However, it is worth mentioning that this has only been tested with XGBoost on GAWAIN, so it is necessary to expand and give justification for this observation in future work. Moreover, the suggestion of missing features given by this addition of embeddings suggests that this classification problem may benefit from the gathering of additional features.

**Table 5.** Top 10 Contributing Features (without embeddings).

| Feature | Contribution(%) |
| --- | --- |
| WHOIS | 13.97 |
| Top-level domain | 4.53 |
| Unique URL Chars count | 4.36 |
| Having @ in URL | 4.29 |
| HTTPS | 4.13 |
| Unique Nums in URL count | 3.75 |
| Location | 2.84 |
| Number of @'s in URL | 2.79 |
| Number of ampersands in URL | 2.74 |

**Table 6.** Top 10 Contributing Features (with PCA 10 components embeddings).

| Feature | Contribution(%) |
| --- | --- |
| URL (Distilbert) PCA embedding 1 | 6.21 |
| Number of ampersands in URL | 2.89 |
| URL (Longformer) PCA embedding 4 | 2.79 |
| DCD Color 1 | 2.74 |
| WHOIS | 2.56 |
| URL (Longformer) PCA embedding 2 | 2.56 |
| HTTPS | 2.47 |
| Top-level domain | 2.33 |
| Is in Alexa's top 1 million | 2.33 |

**Table 7.** Top 10 Contributing Features (with 20 best Chi-squared embeddings).

| Feature | Contribution(%) |
| --- | --- |
| URL (Longformer) embedding 2 | 4.76 |
| Number of Semicolons in URL | 4.14 |
| URL (Distilbert) Embedding 20 | 3.86 |
| Mean of URL (Distilbert) embeddings | 3.69 |
| Mean of URL (Longformer) embeddings | 3.18 |
| WHOIS | 3.02 |
| HTTPS | 2.92 |
| Number of ampersands in URL | 2.82 |
| Mean of feat. selected Longformer embeddings | 2.80 |

We also notice the absence of *Content-based* features altogether from the tables. The information they carry does not seem to be relevant enough to push the accuracy scores higher. One of the main reasons to consider content is to look for suspicious code. However, in our data processing pipeline we have not encountered obfuscated JavaScript code which definitely could play a major role.

## 6. Conclusions and Future Work

In this study, we have provided a realistic outlook on the publicly available data for malicious website detection. Over 25 datasets were analyzed, with only one being relevant for further examination under the constraints we set up beforehand. However, the examined dataset had an issue with the heavily skewed JavaScript features, which resulted in rather optimistic classification scores across multiple models. This led to the creation of a new dataset GAWAIN, where we tried to include as many features as possible that are encountered in the literature and provided access to raw data wherever possible for potentially new feature engineering approaches. As such, our dataset does not display the same type of optimistic results, but seems to give insight into a more realistic application of ML on malicious website detection. We have uploaded our scripts, notebooks and datasets to GitHub and Google Drive [15], with instructions to reproduce the results.

While investigating the influence of particular feature categories, the most dominant one (for XGBoost model) comes from URL embeddings. When all the features where considered together, URL embeddings plus hand-crafted URL features are found to be the most relevant for the classification problem, that is, the lexical category. Other interesting outcome is the absence of content-based features from the most relevant (top-10) features list, while HTML and JavaScript code can contain segments that would be clear indicators of maliciousness, this category has not been as influential as URL-based category. The extracted features for this category seem to be too similar between the two classes to make them useful. Overall, when all the categories where considered together, we were able to achieve the highest prediction accuracy of 84.27% with manual tweaking during feature selection phase. The juxtaposition of multiple categories definitely helped to push the scores higher, as the performance with each category individually was under 80%. With the inclusion of new categories—embeddings from URL, image, and content—the scores were not improved by a large margin, but only slightly. We believe that such minor improvement (less than 1%) can play a crucial role for this particular problem, and in cybersecurity domain in general.

In order to further strengthen the feature category coverage, a potential avenue for further research is to consider fully training neural networks on the raw URLs, images and content, rather then relying on prebuilt models and embeddings. The other improvement is to have a more robust content-based parser, able to correctly identify obfuscated code if

present. We believe that this piece of information is vital, which was not manifested in our preparation of the dataset.

**Author Contributions:** Conceptualization, D.S., A.C., H.S., G.S., X.L.; methodology, A.C. and D.S.; software, A.C.; validation, A.C.; formal analysis, A.C. and D.S.; investigation, A.C.; resources, D.S.; data curation, A.C.; writing—original draft preparation, D.S., A.C., H.S., G.S., X.L.; writing—review and editing, D.S., A.C., H.S., G.S., X.L.; visualization, A.C.; supervision, D.S. and X.L.; project administration, D.S., H.S. and X.L.; funding acquisition, X.L. and D.S. All authors have read and agreed to the published version of the manuscript.

**Funding:** This research was funded by the University of Guelph's Master's of Cybersecurity and Threat Intelligence (MCTI) program of the School of Computer Science (SoCS), Guelph, Ontario, Canada, by Mitacs through the Mitacs Accelerate Program, and by Arctic Wolf Networks, Waterloo, Ontario, Canada.

**Data Availability Statement:** All scripts, Jupyter notebooks and references to datasets can be found on our publicly available GitHub repo https://github.com/AhmadChaiban/Malicious-Website-Feature-Study, accessed on 10 August 2021.

**Conflicts of Interest:** The authors declare no conflict of interest.

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
