# Peer review of "Investigating the Influence of Feature Sources for Malicious Website Detection"

_applsci, doi:10.3390/app12062806_

Round 1

Reviewer 1 Report

Please see the attached PDF: review_applsci-1541718.pdf .

Reviewer 2 Report

This paper investigated the statistical features of malicious websites that enable Machine Learning models to distinguish them from their benign counterparts.
1.The authors need to highlight the main research contribution for this paper.
2.Please check carefully and correct all the typos and format in the revision. 
3.English could be further improved, and all the grammar and composition mistakes should be corrected in the revision

Reviewer 3 Report

The authors propose a machine learning workflow to measure feature sources influences for malicious website detection. This idea sounds interesting, the main highlighted points are as:

1) Abstract section fails to clarify the paper's contribution and innovation when considering the literature gap. It would be interesting to enumerate a brief summarization of the quantitative results. 

2) The problem under study is a contemporary topic nowadays. There are a huge amount of papers (only coming to 2021) about the same topic. Indeed, the authors fail to present a recent literature discussion about the gaps in the literature and place the contributions of the proposal. Instead, the authors claim two core contributions in Section 1 that appear to be already found in the specialized literature. 

3) Section 3 is part of Section 1 (motivation). 

4) Section 4 is part of Section 2 (related works). 

5) Sections 5 and 6 should be merged. Indeed, section 5 is only a setup idea. 

6) From the current version of the paper, the core contribution is limited to Section 6. However, it is presented only a workflow typically already found in the literature. Only the use of a specific application is innovative. Additionally, solutions to detect malicious websites are already found in the literature. Said that the reader has difficulty identifying the contributions. 

7) The results do not support conclusions. 

8) Overall, the idea of the paper is interesting but is missing a better presentation of the proposal, such as a comparison with very recent specialized literature. The contribution is limited to presenting a workflow to evaluate a subset of machine learning models customized for a classification problem. Indeed, this is already found in the literature. 

Round 2

Reviewer 1 Report

The paper has still aspects to reinforce. A more dilated discussion on the bias and on the JS analysis should be desirable. 

Author Response

We appreciate the concern on the JS bias aspect and we agree that it requires reinforcement. We’ve added and made changes which can be found in the last 2 paragraphs of section 3.4. We have added citations to give more insight into our step by step deduction for the bias and JS analysis. 

Reviewer 3 Report

The authors implemented a major revision in the paper. Due to reproducibility issues, the authors are invited to share a public repository with all source code and a README file enumerating how to reproduce results. 

Author Response

Thank you for pointing out that issue, we had appended a link to our GitHub repo in the earlier version as a footnote. We’ve changed this into a reference [39], and have added a sentence in the conclusion that specifically mentions where the information is. Our GitHub repo has a readme that explains the Jupyter notebooks we’ve used that are split into data processing and experimental components.

Round 3

Reviewer 1 Report

Thanks for addressing many of my initial concerns.